# Association of maternal, obstetric, fetal, and neonatal mortality outcomes with Lady Health Worker coverage from a cross-sectional survey of >10,000 households in Gilgit-Baltistan, Pakistan

**Daniel S. Farrar**[1], **Lisa G. Pell**[1], **Yasin Muhammad**[2], **Sher Hafiz Khan**[2], **Zachary Tanner**[1], **Diego G. Bassani**[1,3,4], **Imran Ahmed**[5], **Muhammad Karim**[5], **Falak Madhani**[6,7], **Shariq Paracha**[6], **Masood Ali Khan**[2], **Sajid B. Soofi**[5], **Monica Taljaard**[8,9], **Rachel F. Spitzer**[10,11], **Sarah M. Abu Fadaleh**[1], **Zulfiqar A. Bhutta**[1,3,5,12], **Shaun K. Morris**[1,3,4,13] *

1 Centre for Global Child Health, The Hospital for Sick Children, Toronto, Ontario, Canada, 2 Gilgit Regional Office, Aga Khan Health Service–Pakistan, Gilgit-Baltistan, Pakistan, 3 Temerty Faculty of Medicine, Department of Pediatrics, University of Toronto, Toronto, Ontario, Canada, 4 Dalla Lana School of Public Health, University of Toronto, Toronto, Ontario, Canada, 5 Centre of Excellence in Women and Child Health, Aga Khan University, Karachi, Sindh, Pakistan, 6 Aga Khan Health Service–Pakistan, Karachi, Sindh, Pakistan, 7 Brain and Mind Institute, Aga Khan University, Karachi, Sindh, Pakistan, 8 Ottawa Hospital Research Institute, Clinical Epidemiology Program, Ottawa, Ontario, Canada, 9 School of Epidemiology and Public Health, University of Ottawa, Ottawa, Ontario, Canada, 10 Department of Obstetrics and Gynaecology, University of Toronto, Toronto, Ontario, Canada, 11 Section of Gynecology, The Hospital for Sick Children, Toronto, Ontario, Canada, 12 Institute for Global Health & Development, The Aga Khan University, South-Central Asia & East Africa, Karachi, Pakistan, 13 Division of Infectious Diseases, The Hospital for Sick Children, Toronto, Canada

* shaun.morris@sickkids.ca

**Data Availability Statement:** The data underlying this study contains sensitive and potentially

## Abstract

Pakistan has among the highest rates of maternal, perinatal, and neonatal mortality globally. Many of these deaths are potentially preventable with low-cost, scalable interventions delivered through community-based health worker programs to the most remote communities. We conducted a cross-sectional survey of 10,264 households during the baseline phase of a cluster randomized controlled trial (cRCT) in Gilgit-Baltistan, Pakistan from June–August 2021. The survey was conducted through a stratified, two-stage sampling design with the objective of estimating the neonatal mortality rate (NMR) within the study catchment area, and informing implementation of the cRCT. Study outcomes were self-reported and included neonatal death, stillbirth, health facility delivery, maternal death, postpartum hemorrhage (PPH), and Lady Health Worker (LHW) coverage. Summary statistics (proportions and rates) were weighted according to the sampling design, and mixed-effects Poisson regression was conducted to explore the relationship between LHW coverage and maternal/newborn outcomes. We identified 7,600 women who gave birth in the past five years, among whom 13% reported experiencing PPH. The maternal mortality ratio was 225 maternal deaths per 100,000 live births (95% confidence interval [CI] 137–369). Among 12,376 total births, the stillbirth rate was 41.4 per 1,000 births (95% CI 36.8–46.7) and the perinatal mortality rate was 53.0 per 1,000 births (95% CI 47.6–59.0). Among 11,863 live births, NMR

identifying personal health information. The study participants did not provide informed consent for their information to be made publicly available, and our Research Ethics Board approvals do not permit the authors to deposit the study data into the public domain. However, the study data may be made available to individuals whose secondary data analysis study protocol has been approved by an independent research ethics board. Requests for data access should be made to Dr. Elizabeth Stephenson (Research Ethics Board Chair at The Hospital for Sick Children; elizabeth. stephenson@sickkids.ca) and Dr. Shaun Morris (senior author of the present study; shaun. morris@sickkids.ca). To gain access, individuals will also need to sign a data sharing agreement.

**Funding:** Funding for this study was received from Grand Challenges Canada (authors LGP, DGB, SBS, ZAB and SKM), Aga Khan Foundation Canada (authors LGP, DGB, SBS, ZAB and SKM), and Child Health Evaluative Sciences (author SKM) at the Hospital for Sick Children. The funders had no role in the design, data collection, analysis or interpretation of the study data, or writing or submission of the final report.

**Competing interests:** The authors have declared that no competing interests exist.

was 16.2 per 1,000 live births (95% CI 13.6–19.3) and 65% were delivered at a health facility. LHW home visits were associated with declines in PPH (risk ratio [RR] 0.89 per each additional visit, 95% CI 0.83–0.96) and late neonatal mortality (RR 0.80, 95% CI 0.67–0.97). Intracluster correlation coefficients were also estimated to inform the planning of future trials. The high rates of maternal, perinatal, and neonatal death in Gilgit-Baltistan continue to fall behind targets of the 2030 Sustainable Development Goals.

## Introduction

Global efforts to reduce maternal and newborn deaths are expected to fall short of targets set out in the 2030 Sustainable Development Goals (SDGs) [1–3]. In Pakistan, rates of neonatal death (42 per 1,000 live births) and stillbirth (31 per 1,000 total births) remain second highest amongst all nations [4–6]. Similarly, Pakistan's maternal mortality ratio (186 maternal deaths per 100,000 live births) remains more than double the SDG target (<70 per 100,000) [7]. The majority of maternal, fetal, and newborn deaths in Pakistan are preventable, particularly in remote areas where access to health facilities, skilled birth attendants, and both antenatal and postnatal care are limited [8–10]. For instance, non-sterile delivery environments and lack of umbilical cord care increase the risk of neonatal sepsis, while challenges in identifying early warning signs of infection or hypothermia may result in delayed healthcare seeking and adverse outcomes. To reduce the burden of preventable deaths in the most remote areas, low-cost interventions that can be delivered at-scale through community-based healthcare service models are needed to both prevent serious conditions of newborns and mothers and to provide early warning should these conditions occur [11,12].

Between 2022–2024, a cluster randomized controlled trial (cRCT) in Gilgit-Baltistan, Pakistan aimed to estimate the effect of an integrated newborn care kit on maternal and newborn health outcomes, primarily neonatal mortality (Clinicaltrials.gov NCT04798833). The intervention is designed to be integrated into existing community healthcare infrastructure, and includes tools to be utilized throughout the neonatal period to prevent newborn infection (e.g., clean delivery kit, chlorhexidine gel, sunflower oil emollient), identify and manage hypothermia and fever (ThermoSpot thermoindicator sticker, blanket, and portable heat pack), and teach caregivers on each component's appropriate usage (pictorial guide) [13]. The kit also includes oral misoprostol tablets to prevent postpartum hemorrhage (PPH), the leading cause of maternal death [14]. The cRCT aimed to enroll >27,000 pregnant women from 2022–2024, with both the intervention and control (local standard of care) delivered by Lady Health Workers (LHW). LHWs are government-sponsored community health workers who are trained to provide reproductive, maternal, newborn, and child health (RMNCH) education as well as basic preventive and curative care [15]. Household LHW coverage has previously been associated with improved RMNCH outcomes [15,16].

In this study, we aimed to describe baseline rates of maternal and newborn health outcomes, determine their association with household LHW coverage, and estimate intracluster correlation coefficients in Gilgit-Baltistan, Pakistan to inform the design of future cluster randomized trials in this population.

## Materials and methods

### Study setting and population

Gilgit-Baltistan is the northernmost administrative territory in Pakistan, with an estimated population of 1.8 million people [17]. The geography of the region is defined by the Himalaya

and Karakoram mountain ranges, and the climate is typically arid with extreme fluctuations in seasonal temperatures. Most of the population resides in isolated villages, with access to healthcare services complicated by underdeveloped road infrastructure, harsh winter weather, and frequent landslides. This study was conducted in seven districts of Gilgit-Baltistan including Astore, Diamer, Ghanche, Kharmang, Nagar, Shigar, and Skardu. Data collection in Diamer was limited to the Goharabad, Gonarfarm, and MC Chilas areas. These districts were selected on the basis of anticipated high neonatal mortality rates (NMR), pre-existing LHW coverage, and a lower density of health facilities.

## Study design and procedures

We conducted a cross-sectional, representative household survey between June 14–August 31, 2021. The primary objective of this survey was to collect baseline outcome parameters to inform the cRCT. Specifically, baseline data regarding cluster-level neonatal mortality and health facility deliveries were utilized as inputs for the trial's covariate constrained randomization, and will also be adjusted for in the trial's pre-specified multivariable analyses [13].

The baseline survey was conducted through a two-stage, stratified sampling design (Fig 1). The sampling frame included villages with pre-existing LHW coverage, and selection was stratified by each of the 77 clusters available for randomization in the cRCT. Clusters were defined as Union Councils (or sub-Union Councils), which are geographic subdivisions that are directly involved in local government health administration. To estimate neonatal mortality with equal precision across all clusters, three villages were selected from each cluster with the probability of selection proportional to village population size. Within each selected village, data collection teams comprising three data collectors and one supervisor conducted a standard household listing procedure to enumerate and map occupied households. Forty-eight households per village were selected (or fewer if <48 occupied households were enumerated), using an equal probability systematic selection approach.

We calculated the required sample size of the survey by assuming an NMR of 35 per 1,000 live births, desired precision of 6 per 1,000 live births, and design effect value of 2.03 [5,18]. To achieve this level of precision, we estimated that 7,619 women would need to participate from 11,231 households, assuming a similar response rate and proportion of eligible households as the 2017–18 Pakistan Demographic and Health Survey [5].

## Ethics statement

Verbal informed consent was obtained by study data collectors from one adult representative of the household. At each selected household, data collectors described the study aims, information to be collected, time burden, and confidentiality to at least one adult representative of the household. Family members were encouraged to ask questions, and were informed that participation was completely voluntary. Data collectors then read a consent script directly from the study questionnaire, asking if they would like to participate in the study. The study data collector recorded their own name and study ID on the questionnaire, as well as the household representative's decision to participate as either "Yes, permission given" or "No, permission not given", witnessed by the household representative. In households which contained minors, the consenting adult was a parent, guardian, or other primary caregiver of the minor. The consent discussion was conducted in the family's preferred language. The study was approved by the Research Ethics Board at The Hospital for Sick Children (REB 1000063672), Ethics Review Committee at Aga Khan University (ERC 5250), and the National Bioethics Committee of Pakistan (NBC 580). During the study design phase, our study team also met with representatives of the Gilgit-Baltistan Department of Health to discuss the study

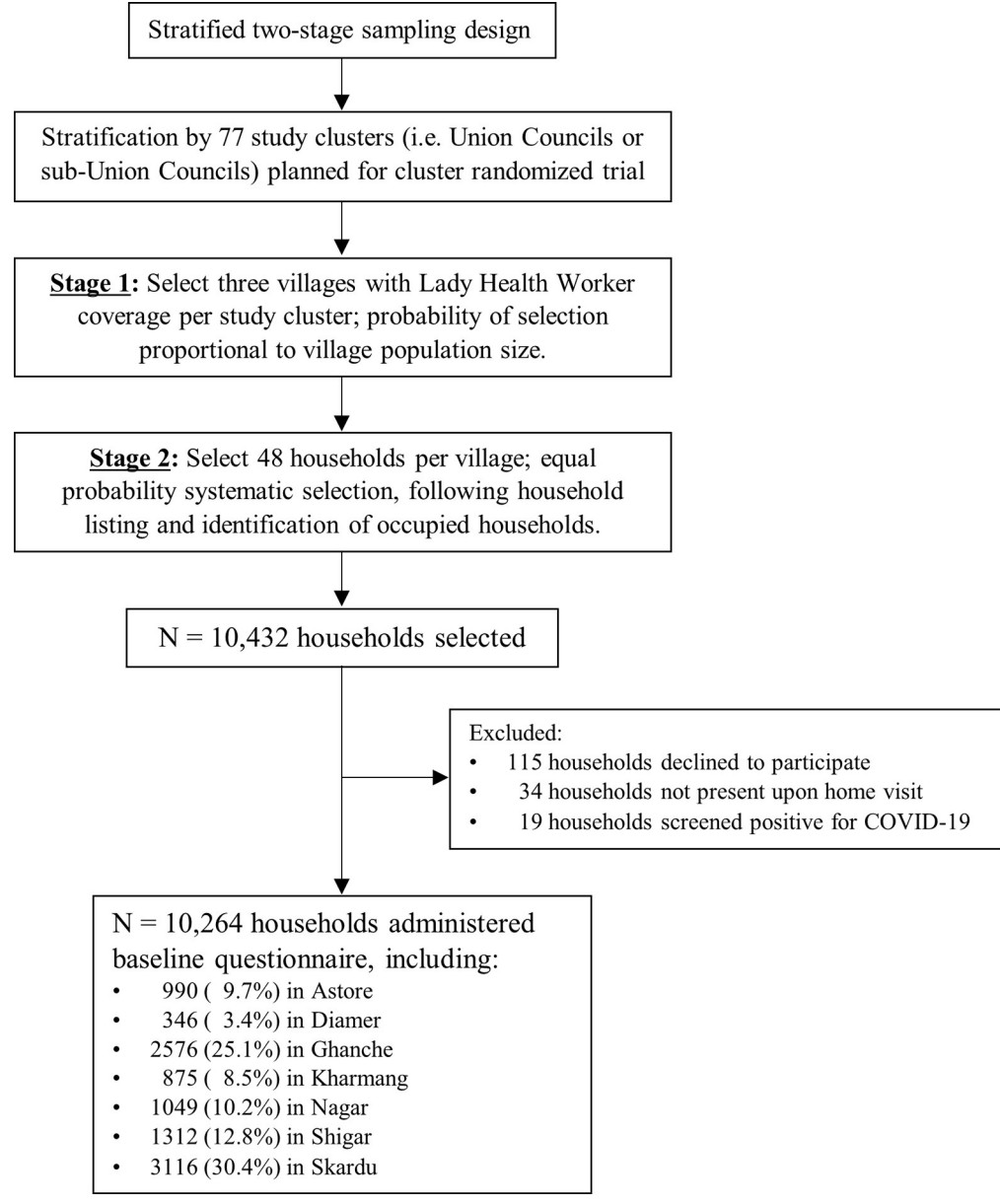

**Fig 1. Study sampling design and flowchart of households included in baseline survey.** Trained data collectors then visited households to administer the study questionnaire; one representative per household was asked to participate and self-report all elements of the questionnaire. The questionnaire was administered primarily in Urdu; however, at least one member of each data collection team was proficient in the local language of each village visited, and translated the study questions as needed into Balti (spoken in Baltistan), Brushiski (spoken in Nagar), or Shina (spoken in Astore). Participants were asked about: a) household reproductive history, including the number of women who gave birth, the number who experienced PPH, and the number of live births and stillbirths delivered; b) live birth history, including the time and place of birth and any neonatal deaths; c) LHW household coverage, including the number of total visits, visits for prenatal care, and visits for postnatal care; and d) household experiences with COVID-19, which will be reported elsewhere. Respondents were asked to report reproductive and live birth outcomes in the past five years, and LHW household coverage in the past 12 months. Finally, quality control officers recontacted approximately 18% of households to verify accuracy of the data collection.

aims and proposed research activities, to ensure the study aligned with local health priorities. Letters of Cooperation were signed by the Gilgit-Baltistan Secretary of Health, following unanimous approval by review committee members. Additional information regarding the ethical,

cultural, and scientific considerations specific to inclusivity in global research is included in the (S1 Checklist).

## Study definitions

In this study, we report on the following maternal, delivery, and newborn health indicators:

- <u>Maternal mortality ratio</u>: The number of women who died due to complications from giving birth (e.g., eclampsia, sepsis, or severe bleeding; occurring within 45 days of delivery), per 100,000 live births.

- <u>Postpartum hemorrhage</u>: The percentage of women who experienced excessive vaginal bleeding within 24 hours of any delivery, among all women who gave birth. Excessive vaginal bleeding was defined as any of the following: blood pooling on the surface of birth environment, healthcare seeking or postnatal blood transfusion for the bleeding, or healthcare worker diagnosis of PPH.

- <u>Crude birth rate</u>: The total number of live births delivered in 2020, per 1,000 total population from sampled households.

- <u>Stillbirth rate</u>: The number of stillbirths, per 1,000 total births.

- <u>Perinatal mortality rate</u>: The number of stillbirths plus the number of neonatal deaths from any cause within the first seven completed days of life, per 1,000 total births.

- <u>Neonatal mortality rate</u>: The number of deaths among live-born children from any cause within the first 28 completed days of life, per 1,000 live births (primary outcome of the cRCT).

- <u>Early neonatal mortality rate</u>: The number of deaths among live-born children from any cause within the first seven completed days of life, per 1,000 live births.

- <u>Late neonatal mortality rate</u>: The number of deaths among live-born children from any cause between 8–28 days of life, per 1,000 live births who survived seven days of life.

- <u>Health facility delivery</u>: The percentage of newborns delivered at any public or private clinic, health unit/center, or hospital, among all live births.

## Statistical analysis

Summary statistics were described using unweighted frequencies, weighted percentages, and weighted continuous statistics (medians and interquartile ranges [IQR], rates with 95% confidence intervals [CI]) among all households and then stratified by district. Household weights were applied to all analyses using Stata's svyset command, and were calculated using templates from the Multiple Indicator Cluster Surveys [19]. These weights accounted for selection at the village and household levels, as well as household non-response. Unadjusted mixed-effects Poisson regression was conducted to assess the association between the number of LHW home visits (prenatal or postnatal; entered as a continuous independent variable) and each RMNCH outcome, using individuals as the unit of analysis. Household ID was included as an additional random effect to account for within-household clustering. We also assessed the association between place of delivery (independent variable) and NMR (dependent variable) using the same method. Lastly, RMNCH indicators were aggregated at the cluster-level, with covariance assessed using Spearman's rank correlation coefficient. All tests of association were conducted at the $\alpha = 0.05$ significance level, and analyses were conducted as complete case

with the unweighted frequency of missing data reported in footnotes. Stata version 17.0 was used for all analyses [20].

To characterize the degree of between-cluster heterogeneity in outcomes, and to aid in the planning of future cluster randomized trials, we calculated intracluster (intraclass) correlation coefficients and coefficients of variation for each RMNCH indicator at the district, Union Council, and village level. Intracluster correlation coefficients were calculated using the large one-way analysis of variance estimator with Stata's loneway command, while coefficients of variation were calculated according to Hayes and Moulton [21].

## Results

In total, 10,264 households agreed to participate in the survey, including 62.7% with ≥1 pregnant women and 62.1% with ≥1 live birth in the past five years (Fig 1, Table 1). Pregnancy histories were collected for 7,600 women, while 12,376 births were recorded including 11,863 live births and 513 stillbirths (Table 2). The median household size was seven occupants (IQR 5–9). Overall, 66.2% of households were visited by LHWs in the past year but this differed by district, ranging from 20.9% in Diamer to 83.5% in Nagar. Among households with any LHW visits, the median number of home visits was five (IQR 4–7). Among households with any pregnant women, 52.3% were visited by LHWs who provided prenatal care, while 77.4% of households with any live birth were visited by LHWs who provided postnatal care.

A history of PPH was self-reported by 13.4% of women who had given birth in the past five years (Table 2). Meanwhile, 30 maternal deaths were reported, corresponding to a maternal mortality ratio of 225 deaths per 100,000 live births (95% CI 137–369 per 100,000). Prenatal LHW home visits were associated with a significant 11% reduction in PPH per each additional

**Table 1. Household characteristics at baseline.**

| Characteristic[1] | Total | District | | | | | | |
|---|---|---|---|---|---|---|---|---|
| | | Astore | Diamer | Ghanche | Kharmang | Nagar | Shigar | Skardu |
| **Enrolled households, N (%)** | 10264 | 990 | 346 | 2576 | 875 | 1049 | 1312 | 3116 |
| **Household size, median (IQR)** | 7 (5–9) | 8 (6–10) | 9 (6–12) | 7 (5–9) | 6 (5–8) | 7 (5–8) | 7 (6–9) | 7 (5–9) |
| **Household outcomes, n (%)** | | | | | | | | |
| Any female aged 15–49 years[2] | 9733 (95.3) | 955 (97.0) | 332 (95.7) | 2388 (92.7) | 816 (92.6) | 1016 (97.6) | 1250 (95.1) | 2976 (95.8) |
| Any pregnancy | 6346 (62.7) | 646 (68.7) | 271 (79.5) | 1567 (60.4) | 471 (53.9) | 651 (61.6) | 857 (62.8) | 1883 (59.2) |
| Any live birth[3] | 6299 (62.1) | 637 (67.1) | 270 (79.4) | 1555 (59.9) | 465 (53.1) | 644 (60.9) | 855 (62.7) | 1873 (59.0) |
| **Any LHW home visits, n (%)[4]** | 6797 (66.2) | 611 (62.9) | 92 (20.9) | 1431 (57.4) | 493 (57.2) | 796 (83.5) | 810 (60.3) | 2564 (82.4) |
| Number of visits, median (IQR)[5] | 5 (4–7) | 6 (5–7) | 4 (3–5) | 6 (4–8) | 3 (2–5) | 5 (4–8) | 6 (5–7) | 5 (4–8) |
| **Any LHW prenatal visits, n / N (%)[6]** | 3321 / 6201 (52.3) | 264 / 636 (37.7) | 19 / 256 (7.8) | 990 / 1536 (65.9) | 249 / 450 (53.1) | 342 / 601 (59.8) | 269 / 853 (27.8) | 1188 / 1869 (59.9) |
| Number of visits, median (IQR)[5] | 3 (2–4) | 2 (2–3) | 1 (1–4) | 3 (2–5) | 2 (1–3) | 4 (3–7) | 3 (2–4) | 3 (2–5) |
| **Any LHW postnatal visits, n / N (%)[7]** | 4858 / 6171 (77.4) | 528 / 631 (86.4) | 20 / 253 (8.2) | 1055 / 1525 (70.2) | 300 / 443 (66.2) | 546 / 600 (91.2) | 788 / 852 (94.6) | 1621 / 1867 (88.1) |
| Number of visits, median (IQR)[5] | 4 (3–6) | 4 (3–5) | 2 (2–4) | 4 (2–6) | 2 (1–3) | 5 (4–7) | 5 (4–6) | 4 (3–5) |

[1]Summary statistics are presented as unweighted counts, weighted percentages, and weighted continuous statistics.

[2]Age and sex not collected in 7 households (4 Diamer, 2 Nagar, 1 Skardu).

[3]Includes 3618 households (36.8% overall) with ≥2 live births.

[4]Not reported for 188 households (10 Astore, 1 Diamer, 37 Ghanche, 47 Kharmang, 78 Nagar, 3 Shigar, 12 Skardu).

[5]Number of visits per household, among households with ≥1 visit.

[6]Among households with any pregnancy.

[7]Among households with any live birth.

**Table 2. Five-year reproductive, delivery, and newborn outcomes at baseline.**

| Characteristic | Total | District | | | | | | |
|---|---|---|---|---|---|---|---|---|
| | | Astore | Diamer | Ghanche | Kharmang | Nagar | Shigar | Skardu |
| **Total women who gave birth, N** | 7600 | 704 | 330 | 1982 | 542 | 743 | 1034 | 2265 |
| **Maternal outcomes** | | | | | | | | |
| Postpartum hemorrhage, n (%) | 974 (13.4) | 139 (19.2) | 24 (7.9) | 260 (12.5) | 48 (8.4) | 78 (10.6) | 220 (22.2) | 205 (11.1) |
| Maternal deaths, n (%) | 30 (0.4) | 2 (0.1) | 2 (0.7) | 6 (0.3) | 4 (0.9) | 1 (0.1) | 8 (0.8) | 7 (0.3) |
| Maternal mortality ratio (95% CI) per 100,000 live births | 225 (137–369) | 75 (15–361) | 356 (38–3246) | 233 (94–576) | 580 (194–1722) | 62 (7–544) | 496 (192–1274) | 181 (72–455) |
| **Total births, N** | 12376 | 1390 | 666 | 2956 | 838 | 1212 | 1693 | 3621 |
| **Birth outcomes** | | | | | | | | |
| Crude birth rate (95% CI), 2020 per 1000 population | 29.3 (27.2–31.5) | 36.0 (30.9–42.0) | 40.4 (27.5–59.0) | 29.5 (25.8–33.8) | 24.7 (21.2–28.8) | 31.9 (27.8–36.6) | 22.1 (17.9–27.3) | 24.4 (22.0–27.1) |
| Stillbirth, n (%) | 513 (4.1) | 92 (4.4) | 26 (4.3) | 130 (4.5) | 25 (3.4) | 26 (2.5) | 69 (4.8) | 145 (4.1) |
| Stillbirth rate (95% CI) per 1000 births | 41.4 (36.8–46.7) | 44.0 (32.4–59.6) | 43.1 (31.2–59.4) | 45.3 (37.5–54.6) | 34.3 (24.2–48.6) | 25.4 (17.1–37.4) | 48.0 (28.4–80.2) | 40.9 (31.0–53.7) |
| **Total live births** | 11863 | 1298 | 640 | 2826 | 813 | 1186 | 1624 | 3476 |
| **Neonatal outcomes** | | | | | | | | |
| Delivered at a health facility, n (%)[1] | 7059 (64.6) | 892 (75.9) | 424 (65.5) | 1434 (51.9) | 376 (46.4) | 962 (82.3) | 780 (48.4) | 2191 (67.8) |
| Neonatal death, n (%)[2] | 204 (1.6) | 13 (1.3) | 4 (0.9) | 73 (2.5) | 7 (0.8) | 23 (1.7) | 36 (24.7) | 48 (1.3) |
| Early neonatal death, n (%) | 151 (77.5) | 13 (100.0) | 3 (62.4) | 49 (66.3) | 7 (100.0) | 19 (84.0) | 24 (77.8) | 36 (76.7) |
| Late neonatal death, n (%) | 53 (22.5) | 0 (0.0) | 1 (37.6) | 24 (33.7) | 0 (0.0) | 4 (16.0) | 12 (22.2) | 12 (23.3) |
| **Mortality rates** | | | | | | | | |
| Perinatal mortality rate (95% CI), per 1000 births[3] | 53.0 (47.6–59.0) | 55.9 (42.3–73.6) | 47.5 (34.6–65.0) | 60.7 (51.4–71.5) | 42.5 (30.7–58.5) | 38.7 (28.3–52.6) | 65.9 (38.7–109.9) | 49.8 (39.6–62.5) |
| Neonatal mortality rate (95% CI), per 1000 live births | 16.2 (13.6–19.3) | 12.6 (7.4–21.4) | 9.2 (2.7–30.7) | 24.9 (18.9–32.8) | 8.5 (4.0–18.1) | 17.0 (11.9–24.2) | 24.7 (14.9–40.9) | 12.9 (8.3–19.8) |
| Early neonatal mortality rate (95% CI), per 1000 live births | 12.4 (10.1–15.2) | 12.5 (7.3–21.2) | 5.6 (2.0–15.6) | 16.3 (11.6–22.8) | 8.4 (3.9–17.9) | 13.8 (9.1–21.0) | 18.9 (9.2–38.6) | 9.7 (6.0–15.6) |
| Late neonatal mortality rate (95% CI), per 1000 live births[4] | 3.7 (2.6–5.3) | — | 3.5 (0.5–22.2) | 8.5 (5.0–14.6) | — | 2.8 (0.9–8.6) | 5.6 (2.5–12.7) | 3.0 (1.4–6.5) |

[1]Reported among live births only; missing for n = 47 live births (17 Ghanche, 3 Kharmang, 14 Nagar, 2 Shigar, 11 Skardu).

[2]Excludes 211 live births who had not yet reached day 28 of life (20 Astore, 11 Diamer, 43 Ghanche, 12 Kharmang, 31 Nagar, 25 Shigar, 69 Skardu).

[3]Perinatal mortality includes all stillbirths plus early neonatal deaths, expressed per 1000 total births.

[4]Denominator excludes live births who died in the first week of life (n = 151) and live newborns who had not yet reached day 28 of life (n = 210).

visit (e.g., one vs. zero visits, two vs. one visit, etc.) (risk ratio [RR] 0.89, 95% CI 0.83–0.96, Fig 2). The association between prenatal LHW home visits and maternal death was RR = 0.79 per each additional visit (95% CI 0.48–1.30).

From all reported births in the past five years, 4.1% were stillborn, corresponding to a stillbirth rate of 41.4 per 1,000 births (95% CI 36.8–46.7). Among all live births, 64.6% were delivered at a health facility and 204 neonatal deaths (1.6%) were reported, including 151 early neonatal deaths (i.e., first seven days of life; 77.5%) and 53 late neonatal deaths (i.e., days 8–28 of life; 22.5%). The NMR was 16.2 per 1,000 live births (95% CI 13.6–19.3), early NMR was 12.4 per 1,000 live births (95% CI 10.1–15.2), and late NMR was 3.7 per 1,000 live births (95% CI 2.6–5.3). Meanwhile, the perinatal mortality rate was 53.0 per 1,000 total births (95% CI 47.6–59.0). Postnatal LHW home visits were associated with a significant 20% reduction in late neonatal mortality per each additional visit (RR 0.80, 95% CI 0.67–0.97, Fig 2). The

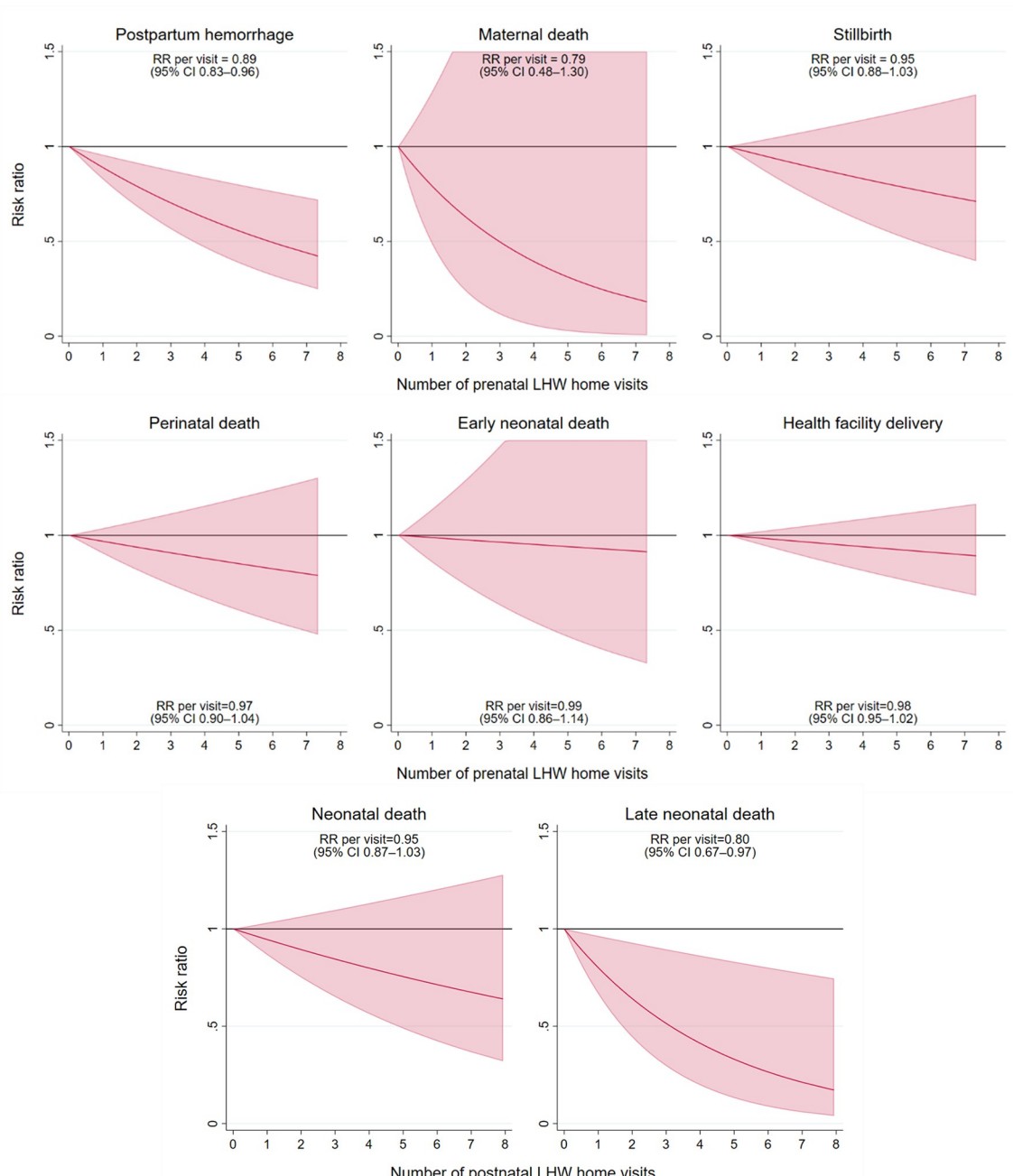

**Fig 2. Associations between maternal, delivery, and newborn outcomes with village-level frequency of Lady Health Worker home visits.** Each line represents the unadjusted risk ratio between prenatal or postnatal LHW visits and the outcome of interest, from an unadjusted mixed effects Poisson regression. Clustering at the Union Council and village levels were accounted for using Stata's svyset command, while within-household clusters was accounted for by including household ID as a random effect in the regression. We assessed for non-linearity by including quadratic terms for the number of LHW visits–interaction terms were removed if not significant using a Wald test. Solid lines indicate a p-value <0.05, while dashed lines indicate p≥0.05.

magnitude of association was smaller between LHW home visits and stillbirth (RR 0.95 per visit, 95% CI 0.88–1.03), neonatal death (RR 0.95 per visit, 95% CI 0.87–1.03), early neonatal death (RR 0.99 per visit, 95% CI 0.86–1.14), perinatal mortality (RR 0.97, 95% CI 0.90–1.04), and health facility delivery (RR 0.98 per visit, 95% CI 0.95–1.02). Across the five-year recall

**Table 3. Intracluster correlation coefficients (ICC) and coefficients of variation (k).**

| Indicator | Cluster unit = District | | | Cluster unit = Union Council | | | Cluster unit = Village | |
|---|---|---|---|---|---|---|---|---|
| | ICC (95% CI) | k | | ICC (95% CI) | k | | ICC (95% CI) | k |
| **Maternal and pregnancy** | | | | | | | | |
| Postpartum hemorrhage | 0.02289 (0–0.05313) | 0.385 | | 0.05130 (0.03251–0.07010) | 0.577 | | 0.07918 (0.05954–0.09882) | 0.717 |
| Maternal death | 0.00021 (0–0.00116) | 0.241 | | 0.00356 (0–0.00792) | 0.990 | | 0.00358 (0–0.00965) | 0.993 |
| **Newborn and delivery** | | | | | | | | |
| Stillbirth | 0.00310 (0–0.00766) | 0.268 | | 0.01402 (0.00757–0.02046) | 0.569 | | 0.02087 (0.01353–0.02820) | 0.695 |
| Perinatal mortality | 0.00227 (0–0.00578) | 0.177 | | 0.01092 (0.00546–0.01638) | 0.442 | | 0.01746 (0.01079–0.02412) | 0.558 |
| Neonatal mortality | 0.00249 (0–0.00631) | 0.388 | | 0.00593 (0.00193–0.00992) | 0.599 | | 0.00651 (0.00176–0.01126) | 0.628 |
| Early neonatal mortality | 0.00074 (0–0.00233) | 0.242 | | 0.00330 (0.00018–0.00641) | 0.513 | | 0.00434 (0.00007–0.00860) | 0.589 |
| Late neonatal mortality | 0.00208 (0–0.00539) | 0.748 | | 0.00463 (0.00102–0.00823) | 1.115 | | 0.00505 (0.00053–0.00956) | 1.165 |
| Health facility delivery | 0.05684 (0–0.12595) | 0.201 | | 0.23694 (0.17627–0.29760) | 0.360 | | 0.26940 (0.22611–0.31269) | 0.384 |

period, the greatest number of neonatal deaths occurred in the month of January (15.9%) while the fewest occurred in May (3.7%; S1 Fig). Neonatal mortality rates were similar between children born at home versus those born in a health facility (S1 Table).

Intracluster correlation coefficients (ICC) and coefficients of variation (k) are shown in Table 3. There was a high degree of clustering for health facility deliveries, with 24% of the variation explained by Union Councils and 27% explained by villages. ICC values were lower for mortality and maternal outcomes. Covariance between all RMNCH indicators, aggregated to the cluster level, are shown in S2 Table. Rates of PPH were significantly correlated with maternal death ($\rho = 0.26$), perinatal mortality ($\rho = 0.29$), neonatal mortality ($\rho = 0.35$), and early neonatal mortality ($\rho = 0.34$).

## Discussion

In this representative, cross-sectional study of >10,000 households in Gilgit-Baltistan, we identified a high burden of maternal, perinatal, and neonatal death. Despite some variability between study districts, six districts exhibited higher maternal mortality ratios (75–580 deaths per 100,000 births) than the 2030 SDG target (<70 deaths per 100,000 live births) while five districts exhibited higher NMR than the SDG target (<12 deaths per 1,000 live births) [2,3,9]. Local indicators of RMNCH including postpartum hemorrhage, maternal death, and perinatal death were also correlated with one another, suggesting some clustering of the poorest outcomes. These areas are most likely to benefit from targeted, community-based interventions such as those delivered in the cluster randomized trial.

We estimated a higher maternal mortality ratio in Gilgit-Baltistan (225 deaths per 100,000 live births, from pregnancy-related causes within 45 days of delivery) than the 2019 Pakistan Maternal Mortality Survey (157 per 100,000 live births, from any cause except accidents or violence, within 42 days of delivery) [7]. Indeed, the highest MMRs occurred in the most remote districts of Diamer (356 per 100,000), Kharmang (580 per 100,000), and Shigar (496 per 100,000), supporting distance to health facilities as a primary contributor to mortality risk [22,23]. We also estimate that approximately one in eight women (13.4%) experienced PPH,

the leading cause of maternal death globally and in Pakistan [14,24]. Few studies have estimated the rate of PPH in Pakistan. However, a 2005–2006 cross-sectional household survey found that 21.3% of women experienced PPH in Khyber Agency, and similarly a randomized trial in Chitral, Khyber Pakhtunkhwa found that 21.9% of women in the control arm experienced PPH [25,26]. Importantly, the diagnosis of PPH is limited in community-based settings where medical attendance of births is not universal, suggesting that data sources relying on self-reported perceived excessive blood loss may overestimate the true burden of PPH. Indeed, a facility-based study of 16 referral healthcare facilities across Pakistan found that only 1.6% of women experienced PPH [27]. Nevertheless, home-based delivery of interventions which reduce the risk of PPH such as oral misoprostol is warranted and has been demonstrated to have efficacy without associated safety concerns [26,28].

Compared to the 2017–18 Pakistan Demographic and Health Survey (DHS), we found lower neonatal mortality (16.2 deaths per 1,000 live births vs. 47 deaths per 1,000 live births in the DHS) but higher perinatal mortality (53.0 deaths per 1,000 total births vs. 43 deaths per 1,000 total births in the DHS) in Gilgit-Baltistan [5]. This is suggestive of a high degree of pregnancy outcome misclassification in our study, whereby early intrapartum neonatal deaths were self-reported as stillbirths, particularly among home births [29,30]. This conclusion is further supported by our high stillbirth estimate (41.4 stillbirths per 1,000 total birth) relative to other Pakistani data sources (30.6 stillbirth per 1,000 total births) [31]. Nevertheless, it remains possible that true declines in neonatal mortality have occurred, alongside those occurring at the national level between the 2012–13 DHS (55 deaths per 1,000 live births) and 2017–18 DHS (42 deaths per 1,000 live births) [5]. Indeed, prior regional estimates from a quasi-experimental study in Gilgit estimated NMRs of 26.0 (in urban areas) and 39.8 (in rural areas) per 1,000 live births from 2002–2003, and declines over the ensuing decades are plausible [32]. Finally, while a third of households lacked any recent LHW coverage, restriction of the sampling frame to villages with anticipated LHW coverage may also have contributed to the lower mortality estimates.

Increased frequency of LHW home visits was associated with significantly lower risk of PPH (11% reduction per additional visit) and late neonatal mortality (20% reduction per additional visit). Moreover, LHW home visits were associated with a 21% reduction in maternal death per additional visit, though this relationship was limited by a small number of maternal deaths and was not sufficiently powered to detect statistical significance. We hypothesize that LHWs are well equipped to identify early warning signs associated with these conditions such as prenatal bleeding, abdominal pain or dizziness, neonatal infection, and hypothermia. This aligns with LHW program policy, whereby LHWs aim to conduct between 4–12 home visits during pregnancy, including for identification of danger signs and health facility referrals [16]. LHWs also aim to complete a first mandatory postnatal visit within the first week of life, plus an additional 1–2 home visits per month to assess the child's nutrition and weight status and deliver first polio vaccinations. In this study, we found a median of four postnatal care visits among households with any LHW coverage, perhaps indicating high motivation among LHWs to provide additional care during this critical period [33]. LHW home visits were not associated with declines in perinatal mortality, reflecting the need for skilled birth attendance to address intrapartum complications. These findings are consistent with a household survey among >26,000 children in Balochistan, Punjab, and Sindh provinces, where LHW covered households experienced reduced post-neonatal mortality and similar rates of neonatal mortality compared to LHW uncovered households between 2015–2017 [34]. Importantly, we were unable to conduct multivariable analysis, and therefore these findings may be confounded by factors such as distance to the nearest health facility and maternal education. Nevertheless, several prior studies have shown improved RMNCH outcomes in LHW covered households, as

well as multiple community-based trials demonstrating effective interventions (e.g., on mortality, nutrition, development) when delivered by LHWs [16,35,36].

The intracluster correlation coefficients and coefficients of variation estimated in this study help to characterize between-cluster variability and may be utilized by researchers for sample size estimation for outcomes related to maternal, perinatal, and neonatal health. Representative baseline data such as these should be used to aid in study planning of comprehensive, high-quality designs.

This study has several limitations. Firstly, the sample frame for the study was limited to areas with anticipated LHW coverage (i.e., to match the delivery of trial interventions) and therefore may exclude more remote areas where RMNCH outcomes are poorest. This may partially explain differences between our study and other household surveys. Moreover, only three Union Councils in Diamer district were included in our study (Goharabad, Gonarfarm, and MC Chilas) and findings may not be representative of the entire district. Second, PPH was collected as a cumulative measure and may not approximate a true incidence rate. Third, place of birth was not collected for stillbirths, and we were therefore unable to further assess differential pregnancy outcome misclassification. Fourth, recall periods for maternal and newborn outcomes (five years) differed from LHW home visits (12 month), and we did not collect data on possible confounders such as distance to the nearest health facility. Fifth, all elements were ascertained by self-report and may have been affected by recall bias, particularly outcomes which used a five-year recall period. Finally, the study recall period included the first 1.5 years of the COVID-19 pandemic, a period when antenatal and perinatal care services may have been less commonly available and when an unknown proportion of the population were infected with SARS-CoV-2. We are unable to comment on what degree these factors affected the maternal and newborn outcomes presented in this study.

## Conclusions

In this study, we confirmed high rates of maternal, perinatal, and neonatal death in Gilgit-Baltistan, Pakistan. Increased frequency of prenatal and postnatal LHW home visits were most strongly associated with declines in postpartum hemorrhage, maternal death, and late neonatal mortality. These findings may be used to plan ongoing and future studies of RMNCH outcomes, and emphasize the need for scalable, low-cost, and community-based interventions to reduce the burden of preventable mortality in the region.

## Supporting information

**S1 Checklist. Inclusivity checklist.** PLOS inclusivity in global research checklist.
(DOCX)

**S1 File. Alternative language abstract.** Alternative language (Urdu) abstract.
(PDF)

**S1 Fig. Seasonality of cumulative monthly newborn deaths from July 2016–August 2021.**
(DOCX)

**S1 Table. Newborn outcomes by place of birth.**
(DOCX)

**S2 Table. Non-parametric (Spearman) correlations between study indicators.**
(DOCX)

## Acknowledgments

We gratefully thank the data collectors, field supervisors, drivers, and other team members for their commitment to this study and for tirelessly administering this survey. We also thank the participating families who welcomed our teams into their homes and shared their experiences.

## Author Contributions

**Conceptualization:** Lisa G. Pell, Sajid B. Soofi, Zulfiqar A. Bhutta, Shaun K. Morris.

**Data curation:** Daniel S. Farrar, Imran Ahmed, Muhammad Karim.

**Formal analysis:** Daniel S. Farrar.

**Funding acquisition:** Lisa G. Pell, Sajid B. Soofi, Zulfiqar A. Bhutta, Shaun K. Morris.

**Investigation:** Daniel S. Farrar, Lisa G. Pell, Yasin Muhammad, Sher Hafiz Khan, Zachary Tanner, Falak Madhani, Shariq Paracha, Masood Ali Khan, Sarah M. Abu Fadaleh, Shaun K. Morris.

**Methodology:** Daniel S. Farrar, Lisa G. Pell, Zachary Tanner, Diego G. Bassani, Monica Taljaard.

**Project administration:** Lisa G. Pell, Yasin Muhammad, Sher Hafiz Khan, Zachary Tanner, Falak Madhani, Shariq Paracha, Masood Ali Khan, Sarah M. Abu Fadaleh, Shaun K. Morris.

**Supervision:** Lisa G. Pell, Sher Hafiz Khan, Falak Madhani, Sajid B. Soofi, Shaun K. Morris.

**Visualization:** Daniel S. Farrar.

**Writing – original draft:** Daniel S. Farrar.

**Writing – review & editing:** Daniel S. Farrar, Lisa G. Pell, Yasin Muhammad, Sher Hafiz Khan, Zachary Tanner, Diego G. Bassani, Imran Ahmed, Muhammad Karim, Falak Madhani, Shariq Paracha, Masood Ali Khan, Sajid B. Soofi, Monica Taljaard, Rachel F. Spitzer, Sarah M. Abu Fadaleh, Zulfiqar A. Bhutta, Shaun K. Morris.

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
