## [Decision Letter · Decision Letter 0]

17 Oct 2023

PGPH-D-23-01815

Association of maternal, obstetric, fetal, and neonatal mortality outcomes with Lady Health Worker coverage from a cross-sectional survey of >10,000 households in Gilgit-Baltistan, Pakistan

Dear Dr. Daniel S Farrar

Thank you for submitting your manuscript to PLOS Global Public Health. After careful consideration, we feel that it has merit but does not fully meet PLOS Global Public Health’s publication criteria as it currently stands. Therefore, we invite you to submit a revised version of the manuscript that addresses the points raised during the review process.

Please read the comments by the reviewer required for revision and answer them carefully in the new version of the manuscript

Please ensure that your decision is justified on PLOS Global Public Health’s publication criteria and not, for example, on novelty or perceived impact.

We look forward to receiving your revised manuscript.

Kind regards,

Mehreen Zaigham

Academic Editor

Journal Requirements:

1. You indicated that you had ethical approval for your study. In your Methods section, please ensure you have also stated whether you obtained consent from parents or guardians of the minors included in the study or whether the research ethics committee or IRB specifically waived the need for their consent.

2. In the ethics statement in the Methods, you have specified that verbal consent was obtained. Please provide additional details regarding how this consent was documented and witnessed, and state whether this was approved by the IRB

3. Please include a complete copy of PLOS’ questionnaire on inclusivity in global research in your revised manuscript. Our policy for research in this area aims to improve transparency in the reporting of research performed outside of researchers’ own country or community. The policy applies to researchers who have travelled to a different country to conduct research, research with Indigenous populations or their lands, and research on cultural artefacts. The questionnaire can also be requested at the journal’s discretion for any other submissions, even if these conditions are not met.  Please find more information on the policy and a link to download a blank copy of the questionnaire here: https://journals.plos.org/globalpublichealth/s/best-practices-in-research-reporting. Please upload a completed version of your questionnaire as Supporting Information when you resubmit your manuscript.”

Additional Editor Comments (if provided):

Reviewers' comments:

Reviewer's Responses to Questions

**Comments to the Author**

1. Does this manuscript meet PLOS Global Public Health’s publication criteria? Is the manuscript technically sound, and do the data support the conclusions? The manuscript must describe methodologically and ethically rigorous research with conclusions that are appropriately drawn based on the data presented.

Reviewer #1: Yes

2. Has the statistical analysis been performed appropriately and rigorously?

Reviewer #1: Yes

3. Have the authors made all data underlying the findings in their manuscript fully available (please refer to the Data Availability Statement at the start of the manuscript PDF file)?

Reviewer #1: Yes

4. Is the manuscript presented in an intelligible fashion and written in standard English?

Reviewer #1: Yes

5. Review Comments to the Author

Reviewer #1: This is a very well written manuscript, enriched with critically important data. I have only one comment:

A big proportion of women who became pregnant and delivered during the pandemic period - might not have ANC PNC services which could have impact on the pregnancy outcome and thereby might impact on the observed result. This can be reported in the manuscript as one limitation aspect.

6. PLOS authors have the option to publish the peer review history of their article (what does this mean?). If published, this will include your full peer review and any attached files.

**Do you want your identity to be public for this peer review?** For information about this choice, including consent withdrawal, please see our Privacy Policy.

Reviewer #1: **Yes: **Rashed Shah

---

## [Editor Report · Decision Letter 1]

28 Dec 2023

PGPH-D-23-01815R1

Association of maternal, obstetric, fetal, and neonatal mortality outcomes with Lady Health Worker coverage from a cross-sectional survey of >10,000 households in Gilgit-Baltistan, Pakistan

Dear Dr. Daniel S Farrar

Thank you for submitting your manuscript to PLOS Global Public Health. After careful consideration, we feel that it has merit but does not fully meet PLOS Global Public Health’s publication criteria as it currently stands. Therefore, we invite you to submit a revised version of the manuscript that addresses the points raised during the review process.

We look forward to receiving your revised manuscript.

Kind regards,

Mehreen Zaigham

Academic Editor

Journal Requirements:

1. You indicated that you had ethical approval for your study. In your Methods section, please ensure you have also stated whether you obtained consent from parents or guardians of the minors included in the study or whether the research ethics committee or IRB specifically waived the need for their consent.

2. In the ethics statement in the Methods, you have specified that verbal consent was obtained. Please provide additional details regarding how this consent was documented and witnessed, and state whether this was approved by the IRB

3. Please include a complete copy of PLOS’ questionnaire on inclusivity in global research in your revised manuscript. Our policy for research in this area aims to improve transparency in the reporting of research performed outside of researchers’ own country or community. The policy applies to researchers who have travelled to a different country to conduct research, research with Indigenous populations or their lands, and research on cultural artefacts. The questionnaire can also be requested at the journal’s discretion for any other submissions, even if these conditions are not met.  Please find more information on the policy and a link to download a blank copy of the questionnaire here: https://journals.plos.org/globalpublichealth/s/best-practices-in-research-reporting. Please upload a completed version of your questionnaire as Supporting Information when you resubmit your manuscript.

Additional Editor Comments (if provided):

Please address the following prior to resubmission of the manuscript.

1) Authors conducting research in other countries or with Indigenous populations are required to complete a copy of PLOS’ questionnaire on inclusivity in global research. The policy applies to researchers who have travelled to a different country to conduct research, research with Indigenous populations or their lands, and research on cultural artefacts. You can find more information on this policy here: https://journals.plos.org/globalpublichealth/s/best-practices-in-research-reporting 

2) This manuscript describes an observational study. Please consider whether this manuscript meets PLOS Global Public Health's guidelines on observational studies involving human subjects https://journals.plos.org/globalpublichealth/s/submission-guidelines. If you would like additional assistance evaluating this manuscript, PLOS Global Public Health Staff Editors and Section Editors have developed a tool https://storage.googleapis.com/genweb.plos.org/RR/EditorResources_CSSAssessment.pdf for you to consider as you determine whether the manuscript should be sent for external peer review. If you feel that the quality of the manuscript does not meet the minimum requirements outlined by this tool, please consider rejecting the manuscript before peer review, ensuring that the decision is justified according to PLOS Global Public Health’s publication criteria. Please contact globalpubhealth@plos.org with any questions or concerns.

---

## [Editor Report · Decision Letter 2]

19 Jan 2024

Association of maternal, obstetric, fetal, and neonatal mortality outcomes with Lady Health Worker coverage from a cross-sectional survey of >10,000 households in Gilgit-Baltistan, Pakistan

PGPH-D-23-01815R2

Dear Daniel,

We are pleased to inform you that your manuscript 'Association of maternal, obstetric, fetal, and neonatal mortality outcomes with Lady Health Worker coverage from a cross-sectional survey of >10,000 households in Gilgit-Baltistan, Pakistan' has been provisionally accepted for publication in PLOS Global Public Health.

Best regards,

Mehreen Zaigham

Academic Editor